# Grade prediction of lesions in cerebral white matter using a convolutional neural network

**Noriaki Takemura**[1], **Yuya Shinkawa**[2], **Kazuo Ishii** [ID][1,3]*

**1** Department of Applied Information Engineering, Faculty of Engineering, Suwa University of Science, Chino, Nagano, Japan, **2** Kurume University Graduate School of Medicine, Kurume, Fukuoka, Japan, **3** Kurume University School of Medicine, Kurume, Fukuoka, Japan

* kazuoishii2014@gmail.com, kishii@rs.sus.ac.jp

**Data Availability Statement:** Additional data to replicate all of the figures, graphs, tables, statistics, and other values is available at doi:10.5061/dryad.007467q as Data files: WM_data.

## Abstract

We established a diagnostic method for cerebral white matter lesions using MRI images and examined the relationship between the MRI images and the medical checkup data. There were approximately 25 MRI images for each patient's head, from the top of the head to near the eyes. To order these images, we defined the unit of axial for convenience. We varied conditions, such as the location and extent of the images to be loaded, into a convolutional neural network model and verified the changes in discrimination performance on the test data. Co-occurrence network diagrams were also used to determine the relationship between the grade of cerebral white matter lesions and the biochemical test items, which were treated as categorical variables, the progression of cerebral white matter lesions, and patient health status. The convolutional neural network showed the highest discrimination performance when the images were loaded into the model with 80 pixels per side, axial from 9 to 15, along with FLAIR and T1-weighted images. The area under the curve for each grade was 0.9814 for grade 0, 0.9800 for grade 1, 0.9905 for grade 2, 0.9977 for grade 3, and 0.9998 for grade 4. In the co-occurrence network diagram, patients with no or mild cerebral white matter lesions, such as grade 0 and grade 1, had near normal blood pressure, whereas grade 2 patients were closer to (isolated) systolic hypertension. This indicates that patients with higher-grade cerebral white matter lesions tend to experience more severe hypertension.

## Introduction

Recently, healthy life expectancy (HALE) has gained attention as Japan's declining birthrate, aging population, and average life expectancy have increased [1, 2]. A reduction in the gap between average life expectancy and HALE through disease and nursing care prevention can prevent the decline in quality of life (QOL), reduce the burden of medical care on individuals, and reduce the burden to social security. Cerebral white matter lesions are associated with dementia [3] and advanced cerebral white matter lesions are a risk factor for cerebrovascular disorders, such as stroke and diabetes [4, 5]. Several studies indicate that there is a significant correlation between the progression of cerebral white matter lesions and a decline in cognitive

**Funding:** The author(s) received no specific funding for this work.

function [6, 7]. Cerebral white matter lesions are ischemic changes caused by cerebral microvascular disease [8] and the greatest risk factor for cerebral white matter lesions is hypertension [4, 9]. Because cerebral white matter lesions progress without any obvious symptoms, progression is often overlooked in daily life, thus it is important to undergo a brain checkup [10]. When cerebral white matter lesions are detected, it is important to take measures to prevent further progression, such as managing blood pressure and improving lifestyle. Brain MRI images may be used to detect cerebral white matter lesions [4, 9]. It is anticipated that automating the detection of cerebral white matter lesions using convolutional neural networks will improve the efficiency of diagnosis.

In a previous study [4], we constructed a model for predicting the presence or absence of cerebral white matter lesions based on logistic regression analysis using data from patients undergoing a brain checkup. We constructed a model to predict the presence or absence of cerebral white matter lesions. Using these data, we established a detection model for cerebral white matter lesions in brain MRI images using a convolutional neural network. In addition, we used a co-occurrence network to examine the relationship between medical checkup data and the other test results.

## Materials and methods

### Subjects

A total of 1146 subjects, who underwent head MRI and blood tests during a comprehensive brain checkup at Kyoju-no-Kai Shin-Takeo Hospital between April 1, 2016, and October 31, 2017, were enrolled. Fig 1 shows the characteristics of the subjects. Of the 1,904 records in the previous study [4], 1,146 individuals were extracted after excluding records that did not contain all three image types, including T1-weighted images, T2-weighted images, and FLAIR images. The number of records by grade was 551 for grade 0 (44.1%), 441 for grade 1 (38.5%), 107 for grade 2 (9.3%), 39 for grade 3 (3.4%), and 8 for grade 4 (0.7%).

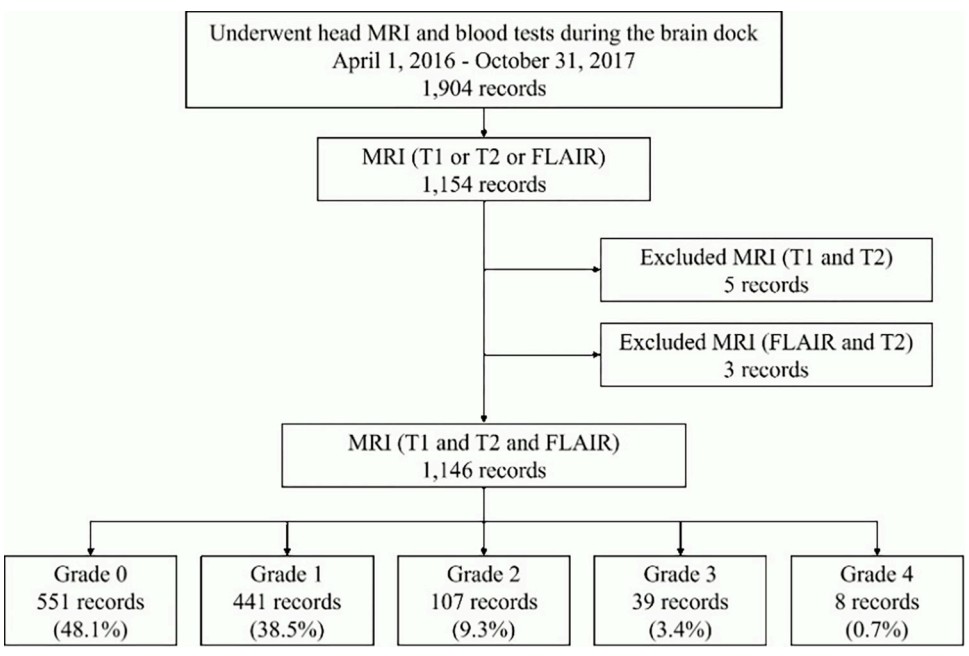

**Fig 1. The characteristics of the subjects.**

The summary statistics for the subjects are listed in Table 1. The patients were divided into 7 groups and graded from 0 to 4 with grades 2 to 4. The means and standard deviations of the biochemical indicators, such as blood tests, taking medication treatment based on a questionnaire, and the percentage with drinking and smoking habits, are shown for each group. In addition, for grades 1 to 4, based on the value of grade 0, a Student's t-test was conducted for continuous variables and a chi-square test was performed for the categorical variables. The significance level was set at 0.05 and Bonferroni's correction was performed.

Each variable listed in Table 1 represents the following items: carotid plaque score (PS [11]), age (age), LDL cholesterol (LDL), HDL cholesterol (HDL), LDL/HDL ratio: quotient of LDL and HDL (LH), triglyceride (TG), hemoglobin A1c (HbA1c), blood glucose level (BS), systolic blood pressure (SBP), diastolic blood pressure (DBP), number of carotid artery plaques (the number of plaques), body mass index (BMI), male (Male), and the presence of visceral steatosis for the determination of metabolic syndrome (metabolic syndrome). In addition, a questionnaire [4] was administered containing the following questions: "Do you have smoking habit, or are you a heavy smoker?" (smoking habit), "Are you taking the following medicines at present: Medication to reduce blood pressure" (medication to reduce pressure), "Are you taking the following medicines at present? Medication to reduce blood sugar or insulin injection" (medication to reduce blood sugar or insulin injection), "Are you taking the following medicines at present? Medication to reduce your level of cholesterol or neutral fat" (medication to reduce a level of cholesterol), "How often do you drink? (such as Sake)" (amount of drinking per day), and "How much do you drink per day?" (drinking habits).

The values for the biochemical and physical test items were used to categorize the diseases into 10 categories, A–G and X–Z, as follows: A: normal blood pressure (SBP<120 and DBP<80), B: normal high blood pressure ((SBP> = 120 and SBP< = 129) and BDP<90), C: high blood pressure ((SBP> = 130 and SBP< = 139) and/or (DBP> = 80 and DBP< = 89)), D: type I hypertension ((SBP> = 140 and SBP< = 159) and/or (DBP> = 90 and DBP< = 99)), E: type II hypertension ((SBP> = 160 and SBP< = 179) and/or (DBP> = 100 and DBP< = 109)), F: type III hypertension (SBP> = 180 and/or DBP> = 110), G: isolated systolic hypertension (ISH) (SBP> = 140 and/or DBP> = 90), X: obesity (BMI> = 25), Y: diabetes (HA1c> = 6.5), Z: dyslipidemia ((LDL> = 140) or (LDL> = 120 and LDL< = 139) or (HDL<40) or (TG> = 150)). However, regarding A–G, if SBP and DBP belonged to different classifications, they were included in the higher classification [12].

Based on the test, the items for which rejection of the null hypothesis was withheld were as follows: PS: grade1, grade2, grade3, age: grade2, grade3, grade4, HbA1c: grade1, grade2, BS: grade3, SBP: grade1, grade2, grade4, DBP: grade1, the number of plaques: grade1, grade2, grade3, smoking habit: grade3, grade4, medication to reduce pressure: grade4, medication to reduce blood sugar or insulin injection: grade4, and normal high blood pressure: grade4.

## Ethical considerations

This study was conducted based on the approval of the ethical review committee of Suwa University of Science. To protect patient privacy, the data was collected with unlinkable anonymization by a third party and was saved in a password-protected storage medium for research use only. Data were accessed for research purposes on 12/06/2019. Additional data to replicate all of the figures, graphs, tables, statistics, and other values is available at doi:10.5061/dryad.007467q as Data files: WM_data. The clinical data used in this study will be available upon request from the corresponding author based on the data usage agreement and considering the patients' right to privacy.

**Table 1. Summary statistics of the subjects.**

| grade | | | all | 0 | 1 | p-value | 2 | p-value | 3 | p-value | 4 | p-value | 2–4 |
|---|---|---|---|---|---|---|---|---|---|---|---|---|---|
| n | | | 1146 | 551 | 441 | | 107 | | 39 | | 8 | | 154 |
| Factor | | | | | | | | | | | | | |
| PS,mean(sd) | | | 0.94(1.8) | 0.53(1.3) | 1.04(1.8) | <0.05* | 1.56(2.2) | <0.05* | 2.92(2.9) | <0.05* | 4.94(4.9) | <0.05* | 2.08(2.7) |
| age,mean(sd) | | | 55.72 (11.7) | 49.62 (10.8) | 59.82 (9.7) | 0.3778 | 64.63(7.7) | <0.05* | 67.82 (4.9) | <0.05* | 71.88 (3.9) | <0.05* | 65.81 (7.2) |
| LDL,mean(sd) | | | 120.52 (30.7) | 119.67 (31.8) | 112.36 (29.8) | 0.3874 | 120.16 (28.8) | 0.8817 | 115.23 (29.4) | 0.3984 | 108.75 (26.0) | 0.3348 | 118.32 (28.8) |
| HDL,mean (sd) | | | 60.25 (15.0) | 59.02 (14.4) | 61.65 (15.5) | 0.3874 | 61.45 (15.9) | 0.1168 | 58.08 (13.4) | 0.693 | 62.13 (12.7) | 0.5445 | 60.63 (15.1) |
| LH,mean(sd) | | | 2.13(0.8) | 2.16(0.8) | 2.12(0.8) | 0.3874 | 2.07(0.7) | 0.3212 | 2.09(0.7) | 0.6186 | 1.83(0.6) | 0.2511 | 2.07(0.7) |
| TG,mean(sd) | | | 110.39 (88.2) | 111.97 (96.2) | 107.03 (79.9) | 0.3778 | 118.96 (91.2) | 0.4878 | 100.51 (49.3) | 0.2031 | 120.38 (51.0) | 0.8053 | 114.36 (80.9) |
| HbA1c,mean(sd) | | | 5.77(0.7) | 5.67(0.6) | 5.84(0.7) | <0.05* | 5.91(0.7) | <0.05* | 5.93(0.6) | <0.05* | 6.06(0.4) | 0.0756 | 5.92(0.7) |
| BS,mean(sd) | | | 104.77 (19.8) | 102.85 (17.6) | 105.89 (21.7) | <0.05* | 107.12 (21.0) | 0.0507 | 111.41 (20.9) | <0.05* | 110.63 (9.0) | 0.2139 | 108.39 (20.5) |
| SBP,mean(sd) | | | 124.22 (18.6) | 120.40 (16.4) | 126.65 (19.2) | <0.05* | 130.00 (20.0) | <0.05* | 132.28 (25.0) | <0.05* | 137.38 (16.7) | <0.05* | 130.96 (21.2) |
| DBP,mean(sd) | | | 74.22 (12.1) | 72.67 (11.5) | 75.62 (12.2) | <0.05* | 76.11 (13.1) | <0.05* | 73.87 (14.9) | 0.6226 | 88.88 (7.6) | <0.05* | 75.79 (13.4) |
| the number of plaque,mean(sd) | | | 0.52(0.9) | 0.30(0.7) | 0.60(1.0) | <0.05* | 0.89(1.1) | <0.05* | 1.44(1.4) | <0.05* | 2.13(1.9) | <0.05* | 1.09(1.3) |
| BMI,mean(sd) | | | 23.19 (3.4) | 23.15 (3.4) | 23.18 (3.3) | 0.8534 | 23.10(3.2) | 0.906 | 24.23 (3.4) | 0.0518 | 23.00 (6.7) | 0.9527 | 23.38 (3.5) |
| Male,n(%) | | | 582 (50.8%) | 443 (80.4%) | 203 (46.0%) | 0.5401 | 43 (40.2%) | 0.2921 | 20 (51.3%) | 0.8105 | 3(37.5%) | 0.2052 | 66 (42.9%) |
| metabolic syndrome, n (%) | no | | 867 (75.7%) | 52(9.4%) | 321 (72.8%) | 0.7485 | 77 (72.0%) | 0.7122 | 21 (55.8%) | 0.1321 | 5(62.5%) | 0.346 | 103 (66.9%) |
| | reserve | | 111 (9.7%) | 52(9.4%) | 42(9.5%) | 1 | 10(9.3%) | 1 | 7(17.9%) | 0.3731 | 0(0.0%) | <0.05* | 17 (11.0%) |
| | yes | | 168 (14.7%) | 56 (10.2%) | 78 (17.7%) | 0.4551 | 20 (18.7%) | 0.3849 | 11 (28.2%) | 0.0596 | 3(37.5%) | <0.05* | 34 (22.1%) |
| smoking habit, n(%) | yes | | 204 (17.8%) | 121 (22.0%) | 65 (14.7%) | 0.5403 | 13 (12.1%) | 0.3393 | 4(10.3%) | 0.2189 | 1(12.5%) | 0.3643 | 18 (11.7%) |
| medication to reduce a blood pressure, n(%) | yes | | 279 (24.3%) | 71 (12.9%) | 139 (31.5%) | 0.07104 | 39 (36.4%) | <0.05* | 24 (61.5%) | <0.05* | 6(75.0%) | <0.05* | 69 (44.8%) |
| medication to reduce a blood sugar and insulin injection, n(%) | yes | | 82(7.2%) | 17(3.1%) | 41(9.3%) | 0.4003 | 17 (15.9%) | 0.0625 | 5(12.8%) | 0.1497 | 2(25.0%) | <0.05* | 24 (15.6%) |
| medication to reduce a level of cholesterol in, n(%) | yes | | 182 (15.9%) | 50(9.1%) | 89 (20.2%) | 0.2253 | 30 (28.0%) | <0.05* | 9(32.1%) | 0.1251 | 4(50.0%) | <0.05* | 43 (27.9%) |
| amount of drinking per day, n (%) (in terms of Sake) | <180ml | | 737 (64.3%) | 320(58%) | 306 (69.4%) | 0.5594 | 75 (70.1%) | 0.5311 | 29 (74.4%) | 0.3793 | 7(87.5%) | 0.1062 | 111 (72.1%) |
| | 180-360ml | | 288 (25.1%) | 160 (29.0%) | 93 (21.1%) | 0.5511 | 26 (24.3%) | 0.7907 | 8(20.5%) | 0.5106 | 1(12.5%) | 0.1036 | 35 (22.7%) |
| | 360-540ml | | 87(7.6%) | 52(9.4%) | 30(6.8%) | 0.911 | 3(2.8%) | 0.4003 | 2(5.1%) | 0.6653 | 0(0.0%) | <0.05* | 5(3.2%) |
| | 540ml< | | 34(3.0%) | 19(3.4%) | 12(2.7%) | 1 | 3(2.8%) | 1 | 0(0.0%) | 0.4286 | 0(0.0%) | 0.4286 | 3(1.9%) |
| drinking habit, n(%) | rarely drink | | 486 (42.4%) | 218 (39.6%) | 200 (45.4%) | 0.7712 | 48 (44.9%) | 0.7907 | 17 (43.6%) | 0.8752 | 3(37.5%) | 0.9959 | 68 (44.2%) |
| | sometimes | | 345 (30.1%) | 165 (29.9%) | 133 (30.2%) | 1 | 37 (34.6%) | 0.8165 | 7(17.9%) | 0.3032 | 3(37.5%) | 0.6319 | 47 (30.5%) |
| | everyday | | 315 (27.5%) | 168 (30.5%) | 108 (24.5%) | 0.7024 | 22 (20.6%) | 0.4304 | 15 (38.5%) | 0.6105 | 2(25.0%) | 0.7401 | 68 (44.2%) |
| A (normal blood pressure),n(%)** | | | 457 (39.9%) | 266 (48.3%) | 153 (34.7%) | 0.367 | 28 (26.2%) | 0.0956 | 9(23.1%) | <0.05* | 1(12.5%) | <0.05* | 38 (24.7%) |

*(Continued)*

Table 1. (Continued)

| grade | all | 0 | 1 | | 2 | | 3 | | 4 | | 2–4 |
|---|---|---|---|---|---|---|---|---|---|---|---|
| n | 1146 | 551 | 441 | | 107 | | 39 | | 8 | | 154 |
| Factor | | | | p-value | | p-value | | p-value | | p-value | |
| B (normal high blood pressure),n(%)** | 193 (16.8%) | 88 (16.0%) | 73 (16.6%) | 1 | 20 (18.7%) | 0.931 | 11 (28.2%) | 0.2716 | 1(12.5%) | 0.8452 | 32 (20.8%) |
| C (high blood pressure),n(%)** | 244 (21.3%) | 119 (21.6%) | 93 (21.1%) | 1 | 23 (21.5%) | 1 | 6(15.4%) | 0.623 | 3(37.5%) | 0.1969 | 32 (20.8%) |
| D (grade I hypertension),n(%)** | 75(6.5%) | 32(5.8%) | 34(7.7%) | 1 | 6(5.6%) | 1 | 2(5.1%) | 1 | 1(12.5%) | 0.4303 | 9(5.8%) |
| E (grare II hypertension),n(%)** | 38(3.3%) | 12(2.2%) | 20(4.5%) | 0.5921 | 2(1.9%) | 1 | 3(7.7%) | 0.3498 | 1(12.5%) | 0.1032 | 6(3.9%) |
| F (grade III hypertension),n(%)** | 12(1.0%) | 3(0.5%) | 4(0.9%) | 1 | 3(2.8%) | 1 | 2(5.1%) | 0.5455 | 0(0.0%) | 1 | 5(3.2%) |
| G ((isolated) systolic hypertention),n(%)** | 127 (11.1%) | 31(5.6%) | 64 (14.5%) | 0.2659 | 25 (23.4%) | <0.05* | 6(15.4%) | 0.2185 | 1(12.5%) | 0.4097 | 32 (20.8%) |
| X (obesity),n(%) | 307 (26.8%) | 145 (26.3%) | 115 (26.1%) | 1 | 29 (27.,1%) | 1 | 17 (43.6%) | 0.193 | 1(12.5%) | 0.1729 | 47 (30.5%) |
| Y (diabetes),n(%) | 92(8.0%) | 25(4.5%) | 46 (10.4%) | 0.4679 | 14 (13.1%) | 0.2556 | 5(12.8%) | 0.2721 | 2(25.0%) | <0.05* | 21 (13.6%) |
| Z (dyslipidemia),n(%) | 690 (59.3%) | 327 (59.3%) | 262 (59.4%) | 1 | 70 (65.4%) | 0.7964 | 17 (43.6%) | 0.3362 | 4(50.0%) | 0.619 | 91 (59.1%) |

Categories A–G and X–Z were shown as follows: A: normal blood pressure (SBP<120 and DBP<80)**, B: normal high blood pressure ((SBP> = 120 and SBP< = 129) and BDP<90))**, C: high blood pressure ((SBP> = 130 and SBP< = 139) and/or (DBP> = 80 and DBP< = 89))**, D: grade I hypertension ((SBP> = 140 and SBP< = 159) and/or (DBP> = 90 and DBP< = 99))**, E: grade II hypertension ((SBP> = 160 and SBP< = 179) and/or (DBP> = 100 and DBP< = 109))**, F: grade III hypertension (SBP> = 180 and/or DBP> = 110)**, G: (isolated) systolic hypertension (SBP> = 140 and/or DBP> = 90)**, X: obesity (BMI> = 25), Y: diabetes (HA1c> = 6.5), Z: dyslipidemia ((LDL> = 140) or (LDL> = 120 and LDL< = 139) or (HDL<40) or (TG> = 150)).

*Bonferroni corrected.

**When SBP and DBP belong to different clessifications, they were included in the higher classification.

## Data and analytical procedure

**Analytical environment, software and library versions.** The specifications of the hardware used in this study were as follows: CPU: Intel Core i9-10850K, RAM: 64.0 GB, GPU: NVIDIA GeForce RTX 3090, OS: Windows 11 Pro, OS version: 22H2. The software was Anaconda3-2021.05. The main libraries and versions were as follows: Python ver3.8.17, Cudatoolkit ver1.3.1, cudnn ver8.2.1, tensorflow-gpu ver2.6.0, keras ver2.6.0, numpy ver1.23.5, scikit-learn, pandas, and seaborn ver0.12.2.

**Grade evaluation of cerebral white matter lesions.** Cerebral white matter lesions were evaluated using MRI images categorized into the following five grades: grade 0, in which no lesions are detected through grade 4, in which most lesions are observed [13], as shown in Table 2 [4, 14].

Fig 2 shows images of patients with and without cerebral white matter lesions. Arrows in the Figure represent cerebral white matter lesions. The upper part of Fig 2 illustrates grade 4

Table 2. Grading the severity of white matter hyperintensities (WMH) [4, 14].

| Deep and subcortical white matter hyperintensity; DSWMH | |
|---|---|
| Grade 0 | Absence |
| Grade 1 | Punctuate foci (< 3 mm in diameter) |
| Grade 2 | Punctuate foci (≥ 3 mm in diameter) |
| Grade 3 | Confluence of foci with unclear boundary |
| Grade 4 | Large confluent areas |

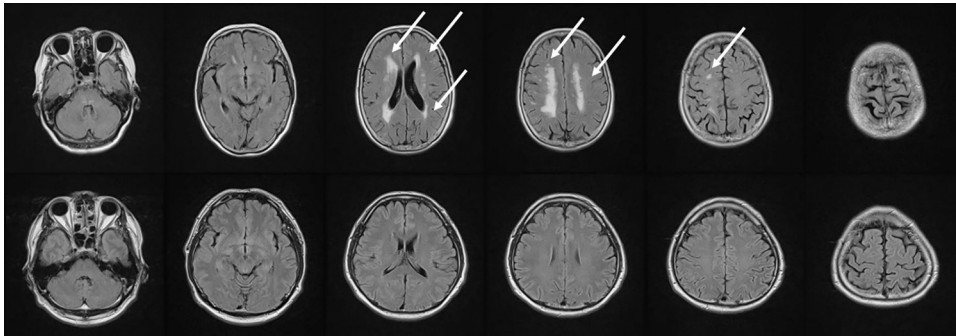

**Fig 2. MRI images of patients with and without typical white matter lesions.** The upper part shows grade 4 and the lower part shows grade 0.

and the lower part represents grade 0. The white areas in the middle of the left and right hemispheres in the third and fourth images from the left in the upper row, and the white spots present in the fifth image from the left in the upper row are cerebral white matter lesions. The larger the white area, the higher the grade.

Three types of MRI imaging methods were used in this study: T1-weighted images, T2-weighted images, and FLAIR images. Fig 3 shows MRI images acquired by the three methods. All images were from the same grade 4 patient and the same area. Arrows in the Figure represent cerebral white matter lesions. The image on the left is a T1-weighted image, the center image is a T2-weighted image, and the image on the right is a FLAIR image. The T2-weighted image depicts water as white with a high signal, whereas the T1-weighted image appears as if the black and white of the T2-weighted image were inverted. The FLAIR image is a T2-weighted image, in which normal water, such as cerebrospinal fluid, is depicted as a low signal and black. It is a useful imaging technique for determining periventricular lesions. Visually, lesions are most clear in the FLAIR images, and lesions are evident to a similar extent in the T2-weighted images [15]. In the T1-weighted images, the lesion area is depicted as a darker gray compared with the surrounding area.

We defined the "axial" and "range" for convenience of analysis. In general, "axial" refers to the axial cross-section in the transverse direction, in which the individual stands upright for CT or MRI imaging. The MRI images in this study were recorded using approximately 25 cross-sectional images for each patient. The "axial" values were assigned in ascending order, with the eyeball side set at 1. The "range" represents the number of "axials" used for learning

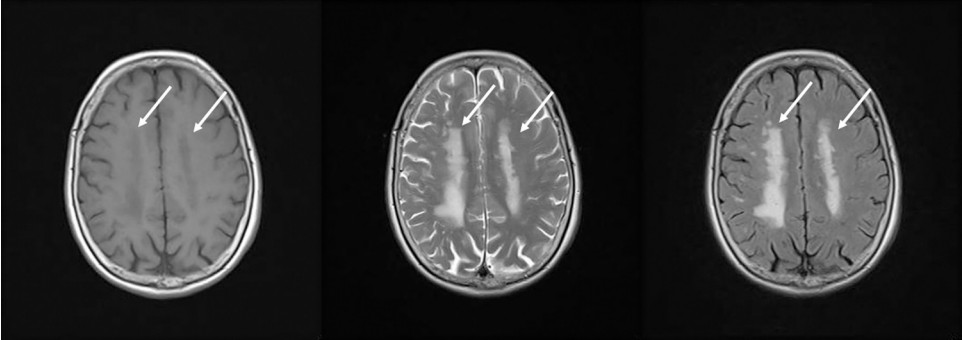

**Fig 3. The three MRI imaging methods.** T1-weighted images (left), T2-weighted images (center), and FLAIR images (right).

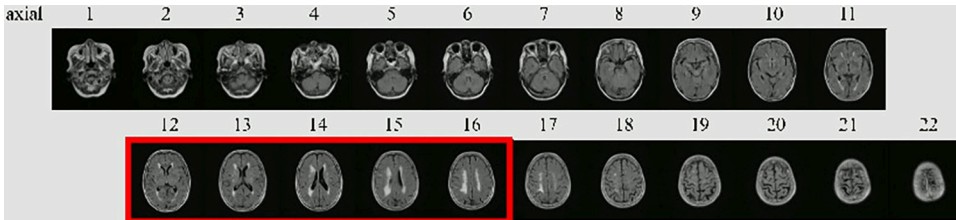

**Fig 4. Definition of axial and range in this study.**

and evaluation of the convolutional neural network. The "axial" represents the part of the approximate 25 images that were extracted and used for learning and evaluation of the convolutional neural network, and the "range" represents the width of the "axial" range used for learning and evaluation. For example, when only the range shown in the red frame in Fig 4 is extracted, the axial is expressed as 12 to 16 and the range is expressed as 5.

**Convolutional neural network.** A convolutional neural network was used to verify changes in the grade discrimination accuracy of the cerebral white matter lesions because of the differences in axial acquisition methods and MRI imaging techniques. The convolutional neural network was constructed using the Keras module, specifically Sequential from keras. models, and Conv2D (convolution layer), MaxPooling2D (pooling layer), Activation, Dropout (dropout layer), Flatten (flatten layer), and Dense (dense layer) from keras.layers. It consists of thirteen layers, as shown in Fig 5 and Table 5, with the following structure: Convolution, Convolution, Pooling, Dropout, Convolution, Convolution, Pooling, Flatten, Dense, Dropout, Dense, Dropout, Dense. Each learning process was performed using the mini-batch method (batch size = 32). The training and testing data were automatically generated by the model_selection.train_test_split method from the sklearn module. Each batch was randomly split into a 3:1 ratio for training data and testing data, respectively. The loss function used in training is categorical_crossentropy from the Keras module. The neural network's output provides probabilities for Grade 0, Grade 1, Grade 2, Grade 3, and Grade 4 through the output layer (type = Dense, output = 5, activation = softmax). The grade with the highest predicted probability is classified as the predicted grade. Each grade evaluation in the ROC curve was conducted using a dataset consisting of a 1:1 ratio between the indicated grade and the other grades, e.g., (Grade 0: Grades 1–4) = 1:1.

Cerebral white matter lesions are often found in the periventricular areas, and lesions do not appear in areas near the eyes or the top of the head, even in patients with severe cerebral white matter lesions. There is a concern that the accuracy may be affected if images in areas where lesions are not observed are used as training data for convolutional neural networks. Therefore, we changed and evaluated the position and number of axes used for learning and evaluation. In addition, the 2019 Brain Checkup Guidelines [14] require a combination of the

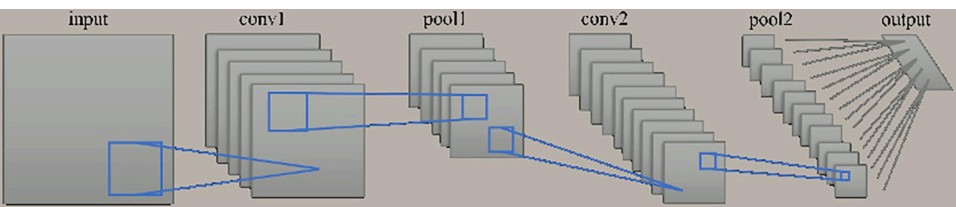

**Fig 5. Overview of the model used in this study.**

**Table 3. Number of images by grade.**

| grade | T1 | T2 | FLAIR | The number of patients |
|---|---|---|---|---|
| 0 | 12739 | 12753 | 12752 | 551 |
| 1 | 10222 | 10223 | 10212 | 441 |
| 2 | 2430 | 2430 | 2430 | 107 |
| 3 | 898 | 898 | 898 | 39 |
| 4 | 181 | 181 | 181 | 8 |
| all | 26470 | 26485 | 26473 | 1146 |

three MRI types for image diagnosis: T1-weighted images, T2-weighted images, and FLAIR images or proton density-weighted images.

We changed and evaluated the performances of the combination of three types of images used for learning and evaluation: T1-weighted images, T2-weighted images, and FLAIR images. For the image analysis data, Table 3 lists the number of images used for learning by grade. This depends on the number of patients in each grade, which is listed in the patient summary statistics of the subjects (Table 1). As the grade increased, the number of images decreased dramatically. In addition, Table 4 shows the number of sheets for each axial direction. Because of the individual differences in head size during MRI imaging, the number of axial images after 23 decreased.

**Table 4. Number of images by axial.**

| axal | T1 | T2 | FLAIR |
|---|---|---|---|
| 1 | 1145 | 1146 | 1146 |
| 2 | 1146 | 1146 | 1145 |
| 3 | 1145 | 1146 | 1146 |
| 4 | 1146 | 1146 | 1145 |
| 5 | 1145 | 1146 | 1146 |
| 6 | 1146 | 1146 | 1145 |
| 7 | 1145 | 1146 | 1146 |
| 8 | 1146 | 1146 | 1145 |
| 9 | 1145 | 1146 | 1146 |
| 10 | 1146 | 1146 | 1145 |
| 11 | 1145 | 1146 | 1146 |
| 12 | 1146 | 1146 | 1145 |
| 13 | 1145 | 1146 | 1146 |
| 14 | 1146 | 1146 | 1145 |
| 15 | 1145 | 1146 | 1146 |
| 16 | 1146 | 1146 | 1145 |
| 17 | 1145 | 1146 | 1146 |
| 18 | 1146 | 1146 | 1145 |
| 19 | 1145 | 1146 | 1146 |
| 20 | 1146 | 1146 | 1145 |
| 21 | 1145 | 1146 | 1146 |
| 22 | 1146 | 1146 | 1145 |
| 23 | 864 | 867 | 867 |
| 24 | 374 | 375 | 374 |
| 25 | 29 | 29 | 29 |
| 26 | 2 | 2 | 2 |
| all | 26470 | 26485 | 26473 |

**Table 5. Overview of the model used in this study.**

| layer | type | | Output Shape | Params |
|---|---|---|---|---|
| 1 | Input | | | |
| 2 | Convolution | Filters = 32, kernel_size = (3,3), Padding = same, activation = relu | (None,80,80,32) | 896 |
| 3 | Convolution | Filters = 32, kernel_size = (3,3),relu | (None,78,78,32) | 9248 |
| 4 | Pooling | Maxpooling, pool_size = (2,2) | (None,39,39,32) | 0 |
| 5 | Dropout | Dropout = 0.25 | (None,39,39,32) | 0 |
| 6 | Convolution | Filters = 32, kernel_size = (3,3), Padding = same, activation = relu | (None,39,39,32) | 9248 |
| 7 | Convolution | Filters = 32, kernel_size = (3,3), Padding = same, activation = relu | (None,39,39,32) | 9248 |
| 8 | Pooling | Maxpooling, pool_size = (2,2) | (None,19,19,32) | 0 |
| 9 | Flatten | | (None,11552) | 0 |
| 10 | Dropout | Dropout = 0.25 | (None,11552) | 0 |
| 11 | Dense | output = 512, activation = relu | (None,512) | 5915136 |
| 12 | Dropout | Dropout = 0.5 | (None,512) | 0 |
| 13 | Dense | output = 5, activation = softmax | (None,5) | 2565 |

An overview of the model used in this study is shown in Fig 5 and Table 5. Two of three-layered structures with two convolutions, one pooling, and three dropout layers were used to create a 13-layer model.

The model was evaluated using the one-versus-rest method to calculate the accuracy, error, TPR (sensitivity) [16], TFR (specificity) [16], PPV (positive predictive value) [17], and NPV (negative predictive value) [17] using a confusion matrix. A ROC curve [18, 19] and the area under the curve (AUC) [20] were used to evaluate the model.

**Convolutional neural network.** Co-occurrence networks are primarily used in natural language processing to visualize the frequency of occurrence of words in a sentence and the relationships between words [21, 22]. To determine the relationship with the clinical test items, we used a co-occurrence network diagram to show the tendency of hypertension, a risk factor for cerebral white matter lesions, and examined its relationship with the progression of cerebral white matter lesions.

Fifteen items, including 10 items consisting of diabetes, dyslipidemia, obesity, and hypertension [normal blood pressure, high-normal blood pressure, high blood pressure, grade I hypertension, grade II hypertension, grade III hypertension, and (isolated) systolic hypertension], and grades 0 to 4 of cerebral white matter lesions, were visualized using a co-occurrence network diagram.

For each patient, the name of their disease among the seven items including diabetes, dyslipidemia, obesity, and hypertension was converted into text in words. Sentences were created for all patients and co-occurrence relationships were formed based on the number of words and the frequency of specific word combinations. The more patients who suffer from a disease, the larger the nodes are and the shorter the edge distances.

## Results

### Convolutional neural network

**Optimization of image size.** We examined the change in "test accuracy" when altering the image size input in the model. The original MRI image was resized from 30x30 to 300x300 pixels using the Image.resize() method from the Python Imaging Library (PIL), and then

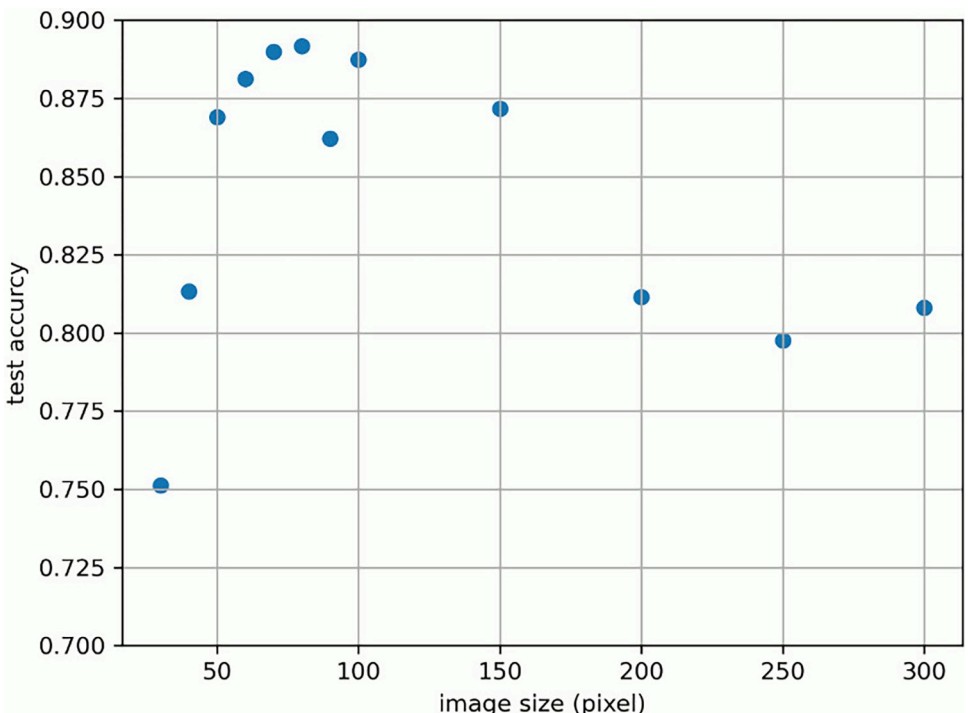

**Fig 6. Test accuracy value when changing image size.**

analyzed. Test accuracy represents the rate of correctly determining the grade of each image using the test data. Fig 6 shows the change in discrimination performance when altering the image size. Only FLAIR images were used and the axial was set from 12 to 15. The horizontal axis is the number of vertical and horizontal pixels of the input image, whereas the vertical axis is the test accuracy. The highest performance was achieved at 80 pixels with a test accuracy of 0.8918. As the image size increased or decreased from 80 pixels as the apex, the discrimination performance tended to gradually decrease. In all subsequent experiments, the image size was set to 80.

**Optimization of "axial" and "range".** Fig 7 shows the change in prediction accuracy for the test data when altering the "axial" used for learning and evaluation. Table 5 provides an excerpt from Fig 7 showing the highest test accuracy value within the same range. The horizontal axis in Fig 7 represents the initial value of the "axial" used, whereas the vertical axis represents the test accuracy. The legend indicates the "range," which is the width of the axial range used. For example, for range = 4 shown in green in Fig 7, when "start axial" is 1, the axes used for learning and evaluation are 1, 2, 3, and 4. When "start axial" is 2, the axes used for learning and evaluation are 2, 3, 4, and 5. Thereafter, when "start axial" is 3, the axes used for learning and evaluation are 4, 5, 6, 4, and so on. The range was moved by 1, such as 5, 6, and 7. The experiments were conducted in a range of 2–9.

Figs 8–15 show Fig 7 divided by "range." As shown in Fig 7, regardless of the number of axes used, the discrimination performance gradually improved until the "start axial" value was approximately 10. From around 12, the discrimination performance decreased while maintaining a certain slope. As shown in Table 6, the highest discrimination performance was obtained when the axial was 9 to 15, the range was 7, and the test accuracy was 0.8958. The range that exhibited the highest discrimination performance was around 7, and the axial range

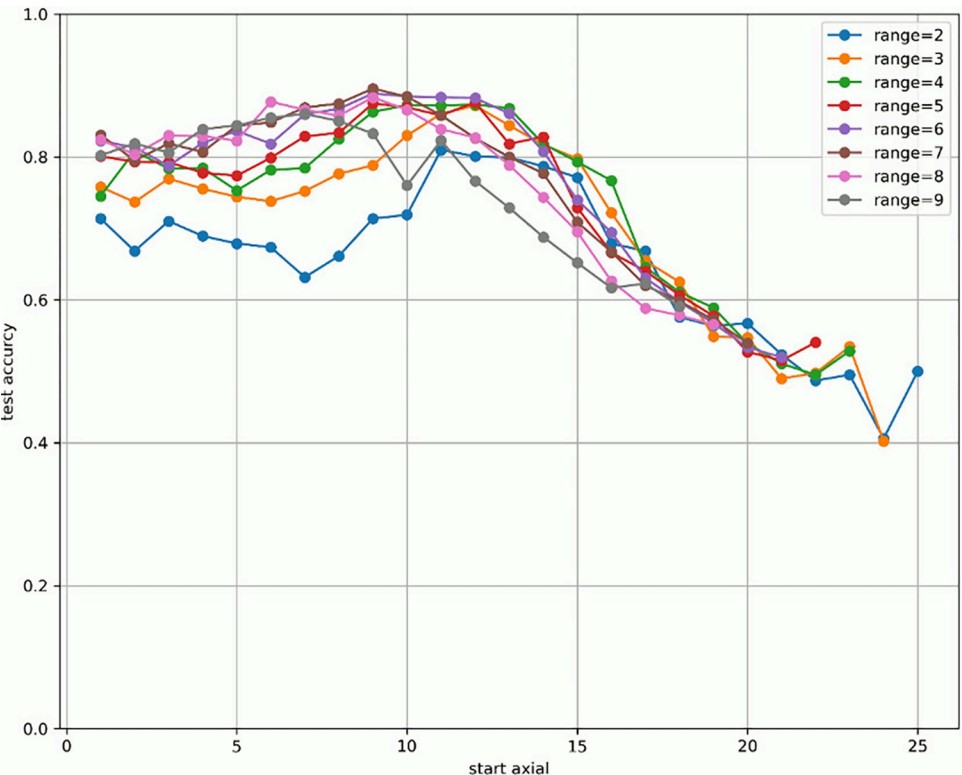

**Fig 7. Test accuracy value when changing range and axial range.**

was approximately 9 to 15. The test accuracy value was low when the axial range was extremely wide or narrow.

**Differences depending on the shooting method.** Fig 16 shows the test accuracy values when changing the combination of images used for learning and evaluation. For all experiments, "axial" was set from 9 to 15 (range = 7). The imaging method showing the highest discrimination performance occurred when FLAIR images and T1-weighted images were used, and the test accuracy was 0.9209.

**Learning curve.** Fig 17 shows the learning curve when the highest discrimination performance was achieved. The upper graph shows the change in accuracy as learning progresses. The horizontal axis represents the number of epochs and the vertical axis indicates accuracy. The lower graph shows the change in loss over the course of learning, with the horizontal axis representing the number of epochs and the vertical axis indicating the loss. In both figures, the blue line indicates the performance on the training data, and the orange solid line represents the performance on the test data. In terms of both accuracy and loss, performance improved for both the training data and test data as learning progressed, but only in loss, there was a slight tendency for overfitting.

**Evaluation by grade.** Fig 18 shows a heat map representation of the confusion matrix by grade. The horizontal axis indicates the grade predicted by the model for the test data, whereas the vertical axis represents the actual grade; that is, the correct answer label. Normally, a confusion matrix for binary classification is represented by a 2x2 confusion matrix of TP (true positive), TN (true negative), FP (false positive), and FN (false negative) values; however, because this task is a multiclass classification that predicts 5 grades, it is represented by a 5x5 confusion matrix.

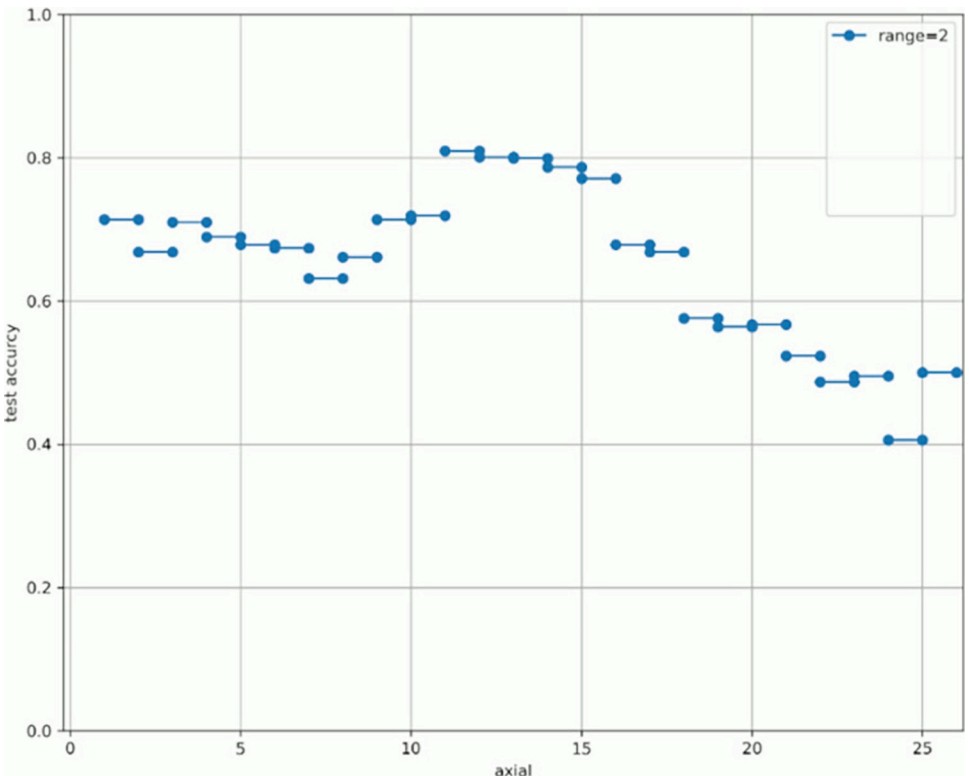

**Fig 8. Range = 2.**

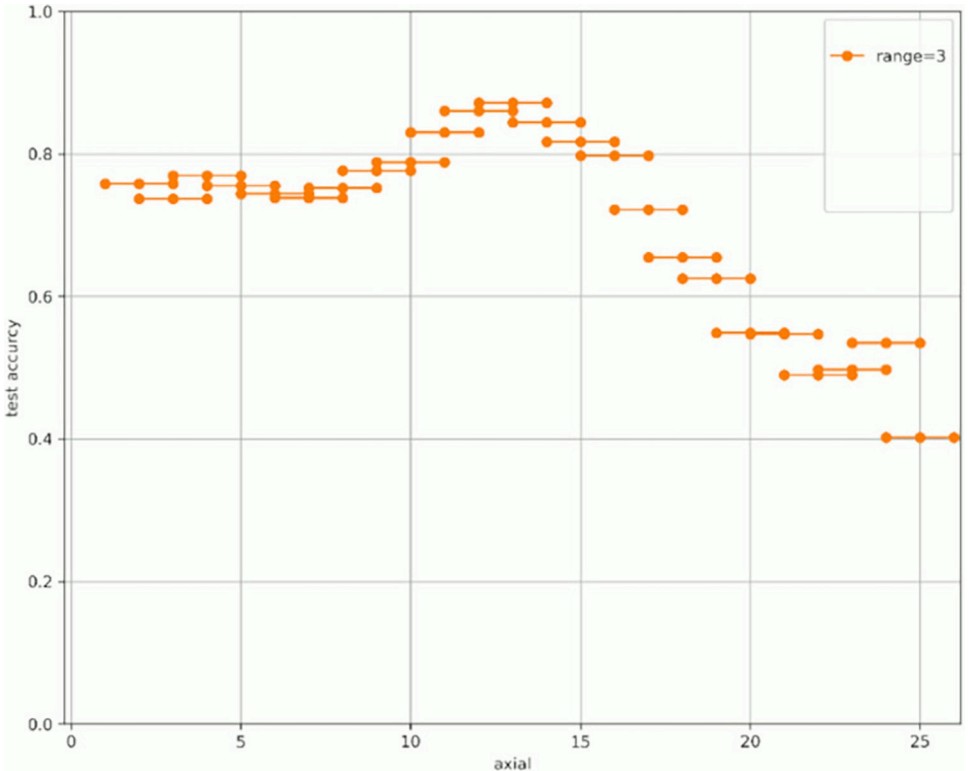

**Fig 9. Range = 3.**

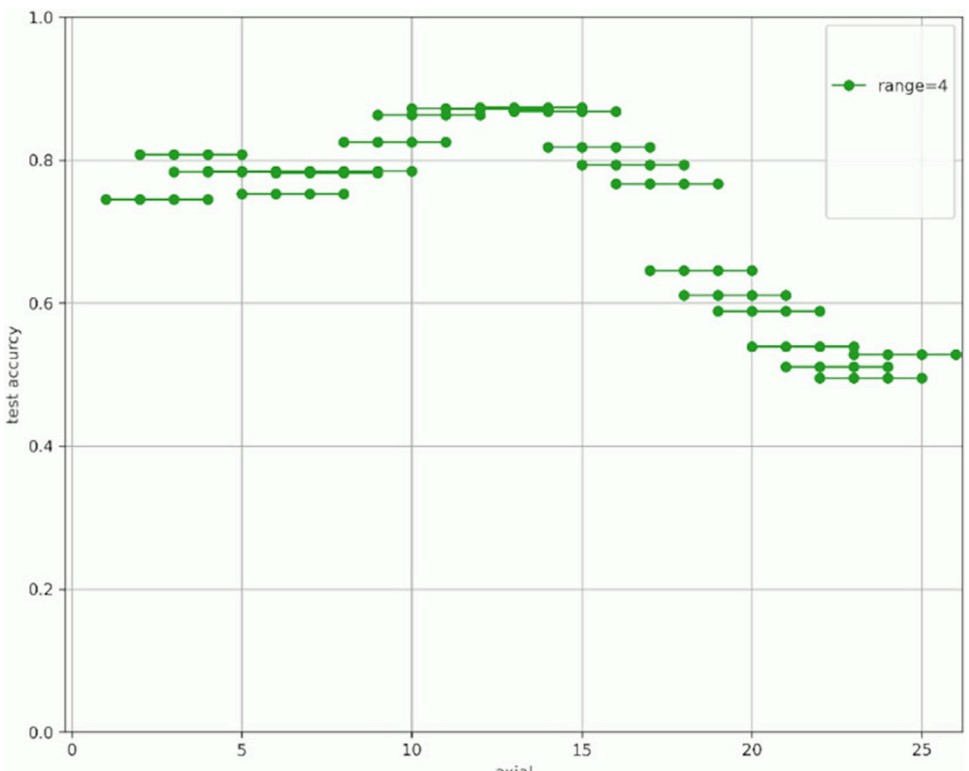

**Fig 10. Range = 4.**

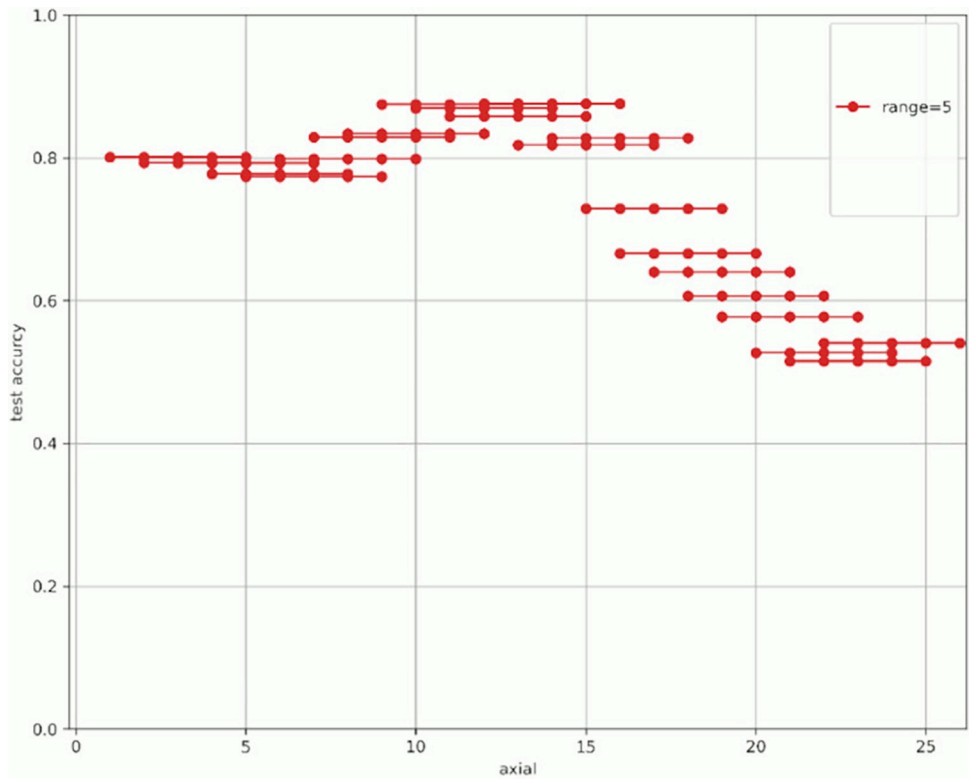

**Fig 11. Range = 5.**

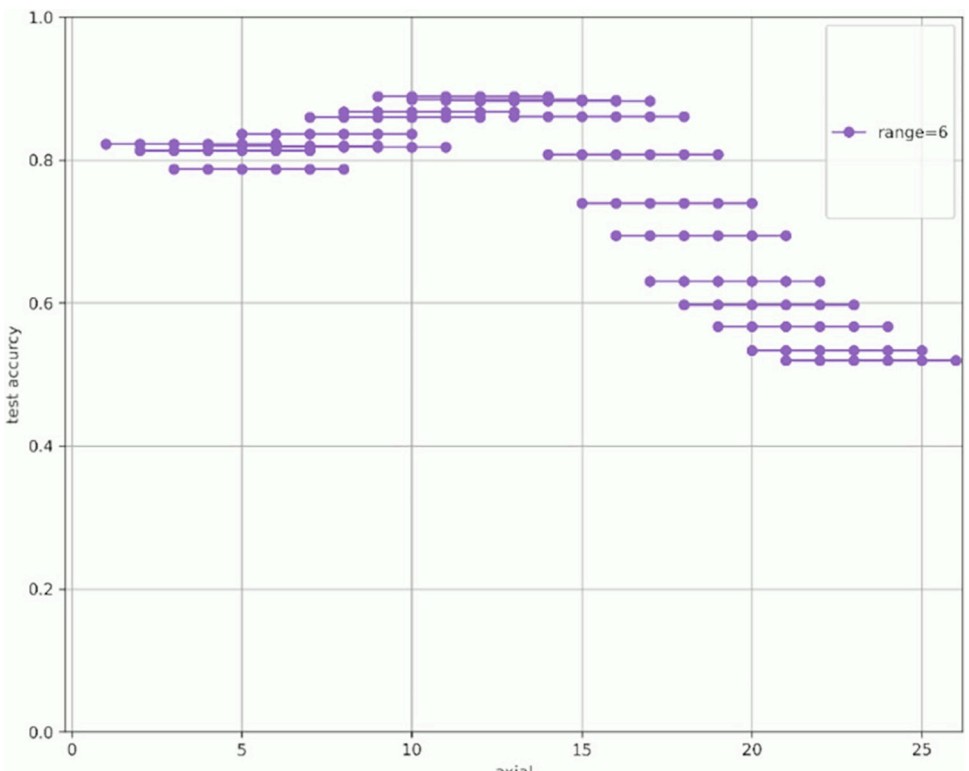

**Fig 12. Range = 6.**

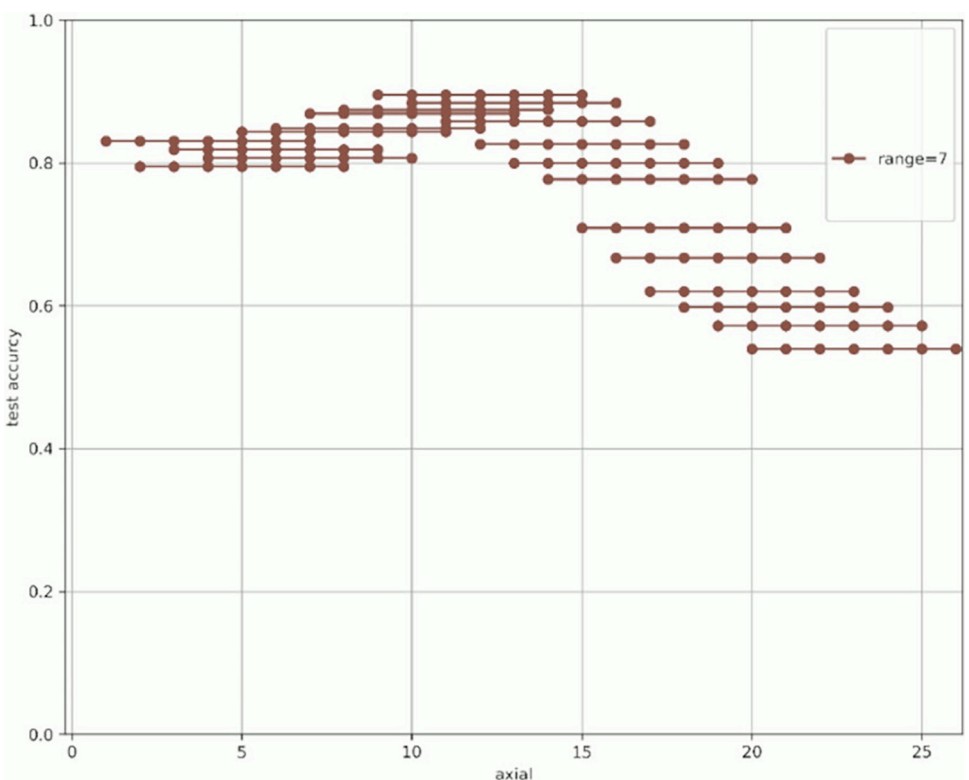

**Fig 13. Range = 7.**

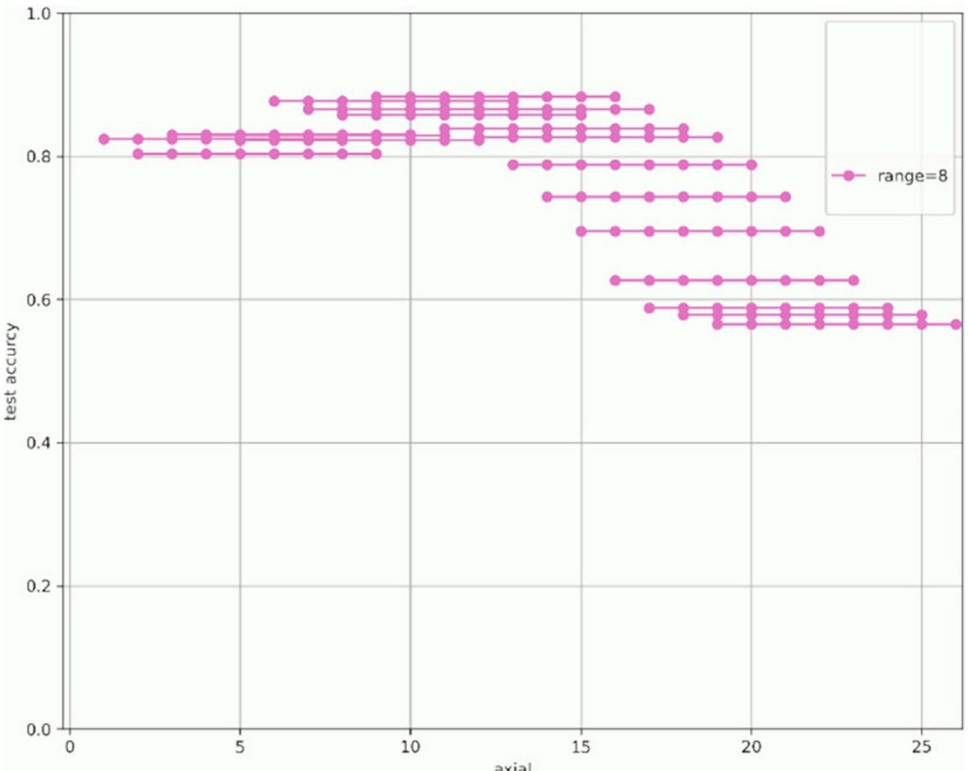

**Fig 14. Range = 8.**

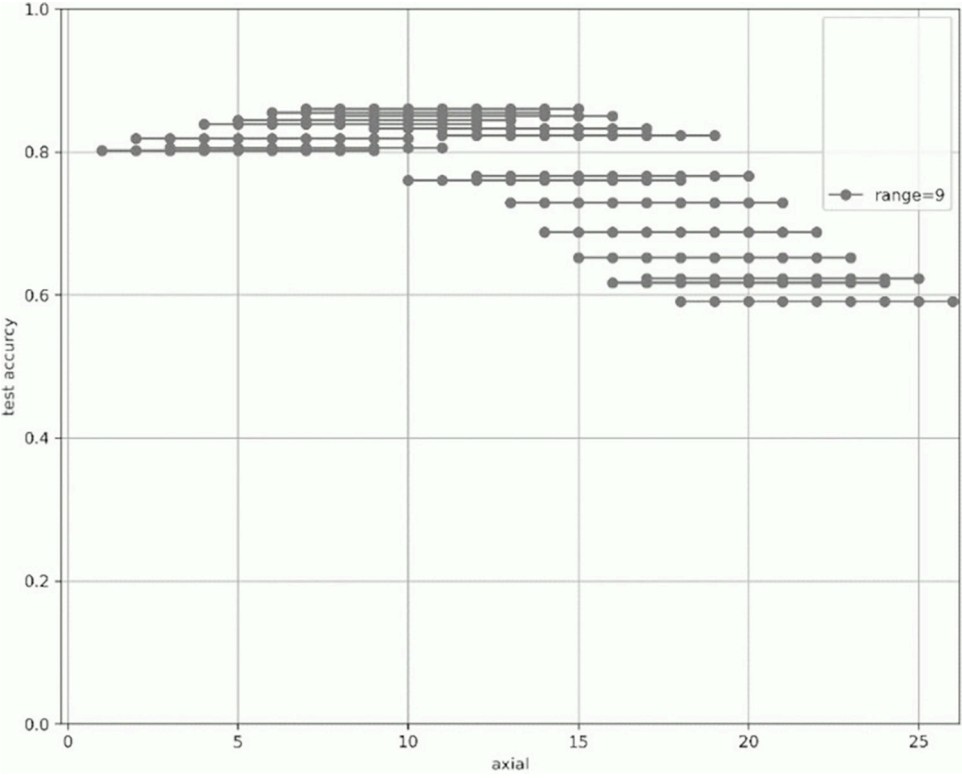

**Fig 15. Range = 9.**

**Table 6. The axial values that showed the highest test accuracy by range based on Fig 7.**

| Start | end | range | test_accuracy |
|---|---|---|---|
| 9 | 15 | 7 | 0.8958 |
| 9 | 14 | 6 | 0.8889 |
| 9 | 16 | 8 | 0.8835 |
| 12 | 16 | 5 | 0.8764 |
| 12 | 15 | 4 | 0.8735 |
| 12 | 14 | 3 | 0.8179 |
| 7 | 15 | 9 | 0.8604 |
| 11 | 12 | 2 | 0.8098 |

To evaluate performance by grade, we used the One vs Rest method, which evaluates performance by binary classification. This considers only the grade of interest as a positive example and the others as negative examples. For example, to evaluate the discrimination performance of grade 1, one can create a general binary classification confusion matrix, such as Fig 19, by dividing the classes into grade 1 and the other grades as shown in Fig 20.

Table 7 shows the results of accuracy (accuracy), error rate (error), sensitivity (TPR), specificity (TFR), positive predictive value (PPV), and negative predictive value (NPV) by grade using this evaluation method. Fig 21 shows ROC curves plotted by grade, with sensitivity on the vertical axis and 1-specificity, or false positive rate, on the horizontal axis. The results

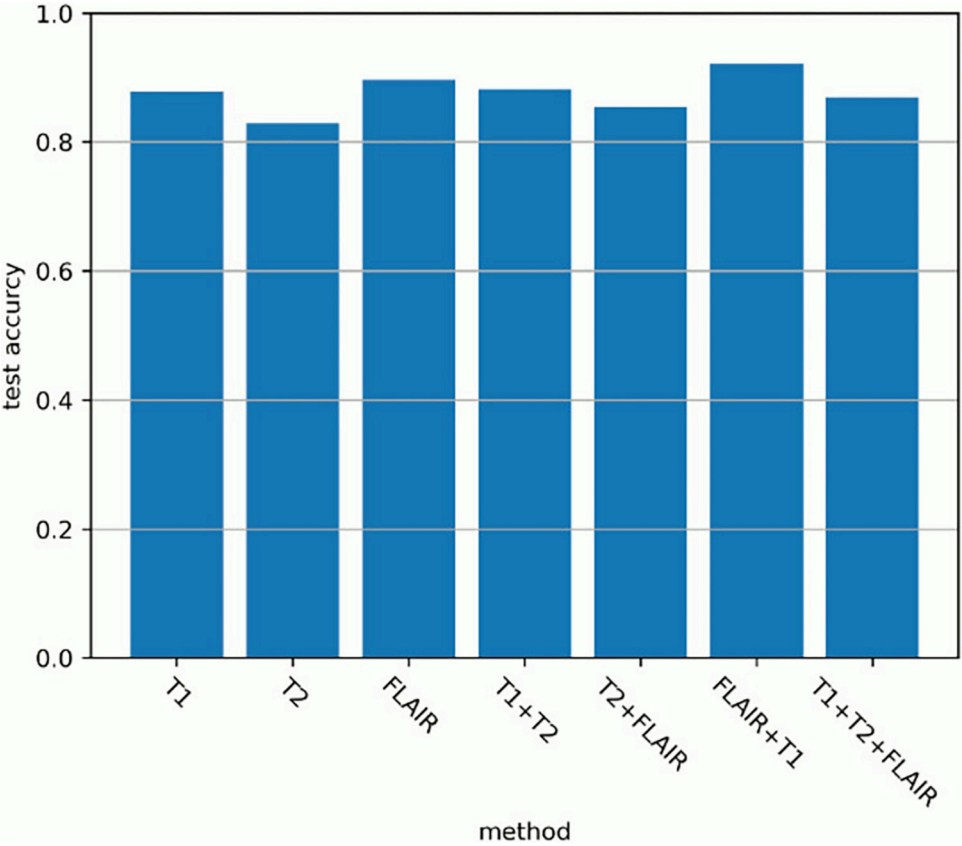

**Fig 16. Changes in discrimination performance when changing the combination of the shooting methods.**

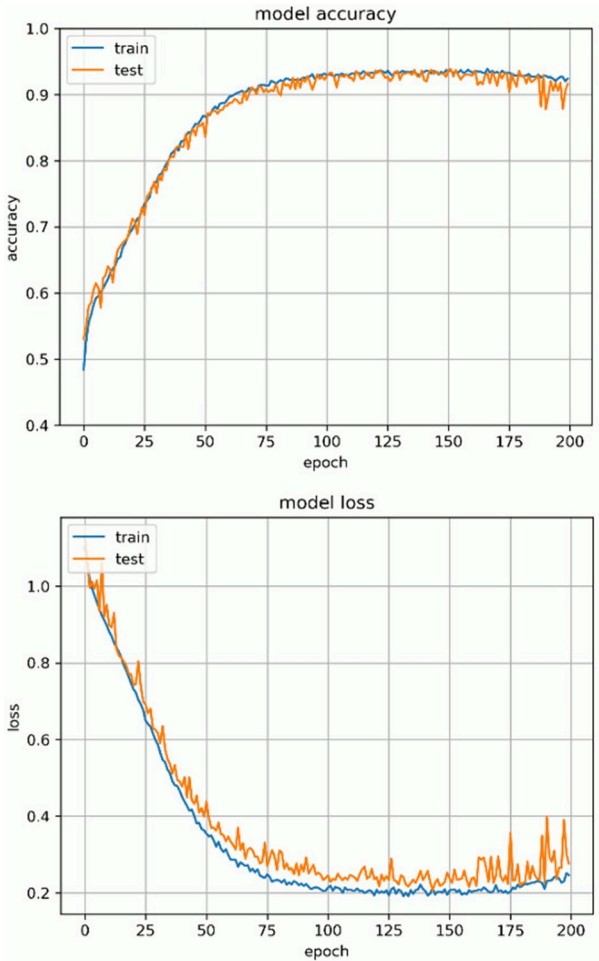

**Fig 17. Learning curve.**

indicated that grade 0 and grade 1 had almost the same performance; however, as the grade increased, the discrimination performance improved. The performance of the model was evaluated based on the AUC, which represents the area under the ROC curve. The AUC of each grade was 0.9814 for grade0, 0.9800 for grade1, 0.9905 for grade2, 0.9977 for grade3, and 0.9998 for grade4.

**Co-occurrence network diagram.** Fig 22 shows the results of visualizing the five grades of cerebral white matter lesions, the seven categories of hypertension classified according to severity, and the 15 categories of obesity, diabetes, and dyslipidemia based on a co-occurrence network diagram.

The size and color density of the nodes indicate the number of samples, whereas the distance between nodes represents the strength of the co-occurrence relationship. Therefore, the closer the distance between nodes, the more closely related they are to having the two diseases. Grade 0 and 1 are close to nodes representing blood pressure within or close to the normal range, such as normal blood pressure and normal high blood pressure, whereas grade 2 is far from these blood pressure nodes (normal blood pressure and normal high blood pressure) and was close to (isolated) systolic hypertension, which is more likely to be observed in the elderly. This indicates that grade 0 and 1 patients are more likely to have normal or near normal blood

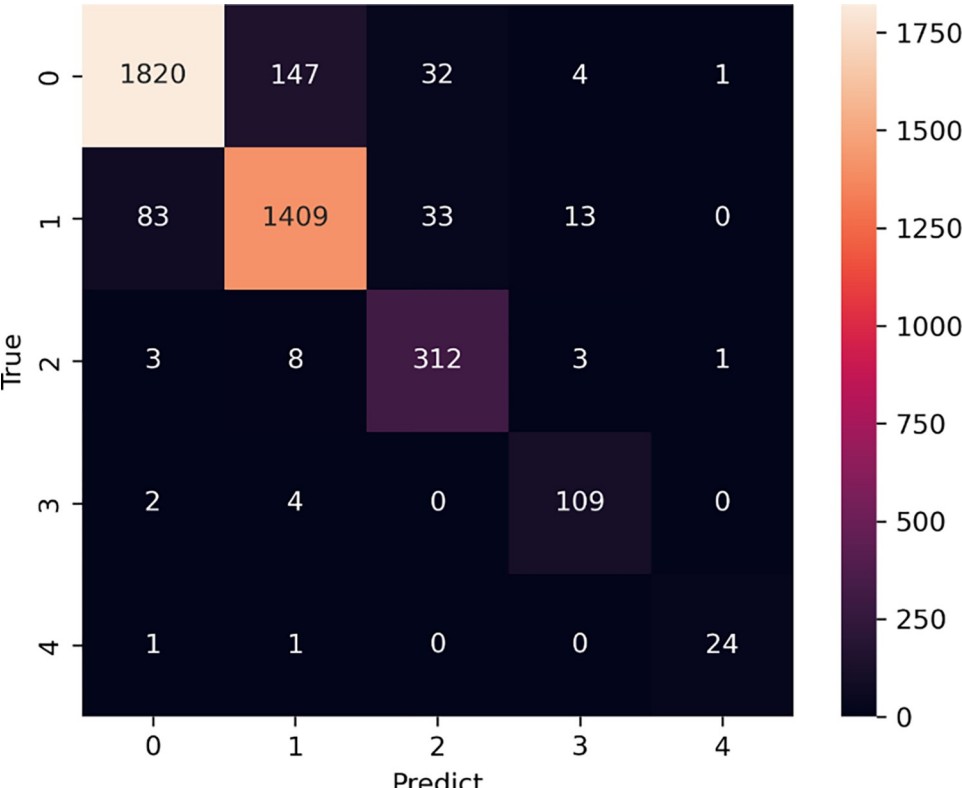

**Fig 18. Heat map representation of the confusion matrix.**

pressure, whereas grade 2 patients are more likely to have hypertension. Thus, patients with more advanced cerebral white matter lesions had more severe hypertension.

## Discussion

For grade prediction using convolutional neural networks, the accuracy tends to increase as the range value increases because the number of images used for learning increases; however, if the range is increased too much, the training data will include images from the eyeball side and the parietal side, where cerebral white matter lesions are not detected, so the accuracy is thought to have decreased.

For grade prediction using a convolutional neural network for each imaging method, the accuracy was higher when using FLAIR images because it is easier to visually identify cerebral white matter lesions and easier to extract features by convolution. It was also easy to visually identify lesions in T2-weighted images, but the peripheral parts of the cerebrum are drawn white as high-intensity areas. Therefore, it is difficult to make a clear distinction between lesions. This is why the discrimination performance using T2-weighted images was low.

Because the number of samples of grade 3 (or higher) was small in the co-occurrence network diagram, no relationships could be drawn, and they were not drawn as clear nodes. Therefore, it was not possible to fully examine the relationship between patients with severe cerebral white matter lesions and health checkup items. Regarding the relationship between test items in health checkup data, we intend to examine the relationship between cerebral white matter lesions and their risk factors by considering the use of drugs, such as antihypertensive drugs, and the relationship with patient history, such as cerebral infarction.

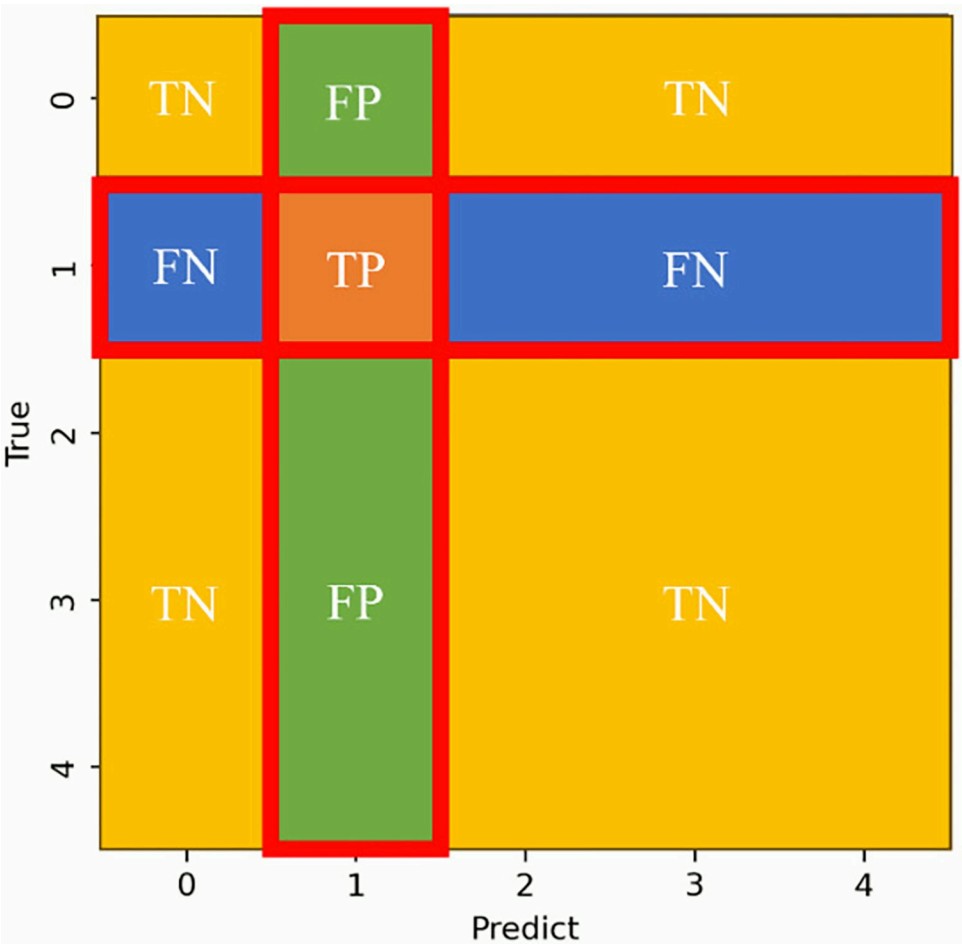

**Fig 19. Confusion matrix for binary classification.**

The grade distribution was as follows: Grade 0: 551 (48.1%), Grade 1: 441 (38.5%), Grade 2: 107 (9.3%), Grade 3: 39 (3.4%), and Grade 4: 8 (0.7%) among 1,146 subjects. This likely reflects the actual grade distribution in this hospital and region. Normally, a small number of patients show high grade variance in the grading. The results for Grades 3 and 4 may indicate less

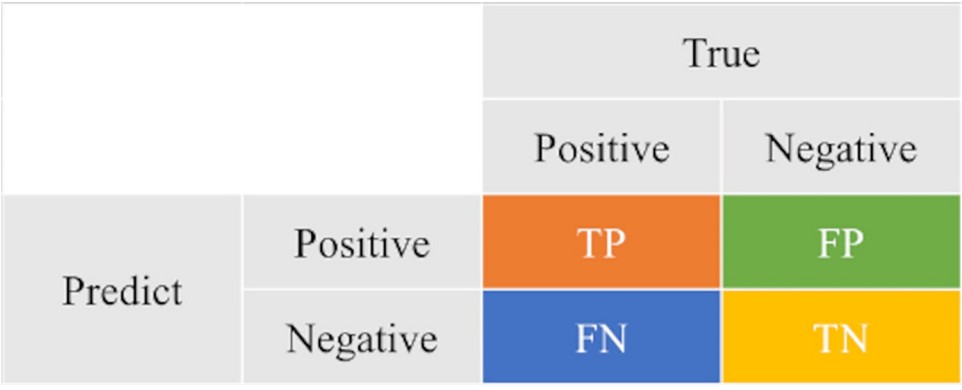

**Fig 20. Conceptual diagram of the One vs Rest method.**

**Table 7. Discrimination performance by grade.**

| grade | accuracy | Error | TPR | TFR | PPV | NPV |
|---|---|---|---|---|---|---|
| grade 0 | 93.39% | 6.81% | 98.82% | 95.56% | 95.34% | 91.24% |
| grade 1 | 92.79% | 7.21% | 91.61% | 93.53% | 89.90% | 94.72% |
| grade 2 | 98.00% | 2.00% | 95.41% | 98.24% | 82.76% | 99.59% |
| grade 3 | 99.35% | 0.65% | 94.78% | 99.49% | 84.50% | 99.85% |
| grade 4 | 99.90% | 0.10% | 92.31% | 99.95% | 92.31% | 99.95% |

generalizability, probably due to overfitting. Therefore, there may be some limitations in the higher grades within this grading model. However, the model might still be practical for clinical use in screening, as physicians can perform detailed inspections to verify diagnoses. As the number of cases accumulates, the model should be able to achieve greater generalizability.

Since machine learning is typically data-driven, the proposed model may be practical for clinical use across different populations and MRI devices. However, there are some challenges in maintaining generalizability. The grading quality should be upheld by continuously updating the model with an appropriate training dataset, as the quality of grading depends on the quality of the training data. To apply the proposed model to different populations or MRI devices, ML users must ensure and maintain grading quality through techniques like transfer learning and fine-tuning, while also evaluating the quality of the data grading. Therefore, the proposed model is neither final nor perfect in terms of generalizability.

We can enhance the model's interpretability by using causal inference approaches, such as logistic regression analysis and Bayesian network models, instead of a co-occurrence network. A directed network, as opposed to an undirected network, can provide some interpretability to

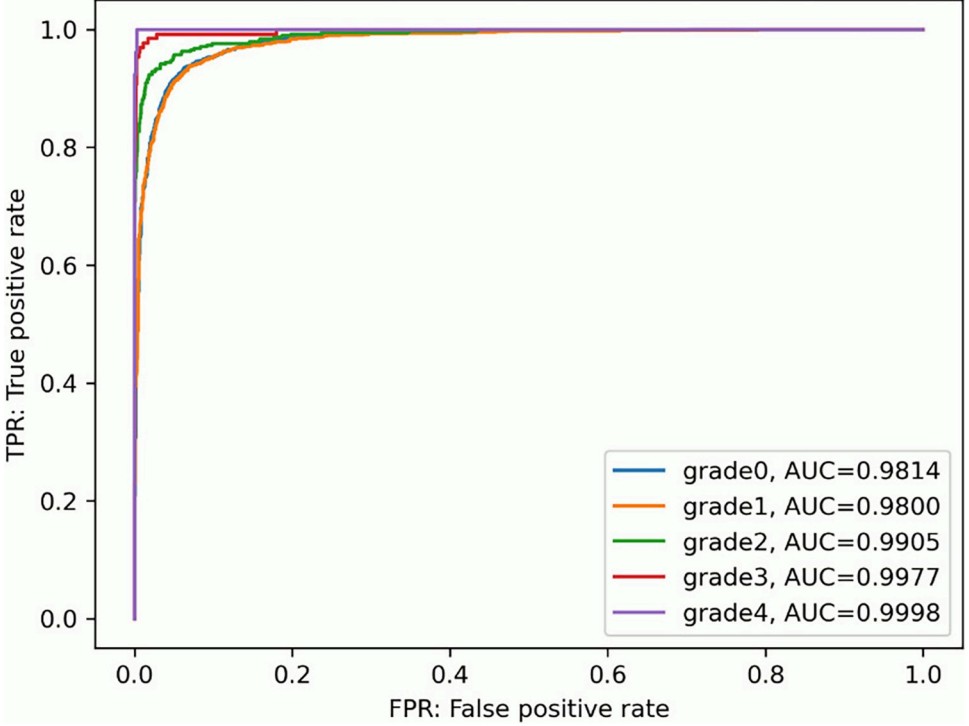

**Fig 21. ROC curve.**

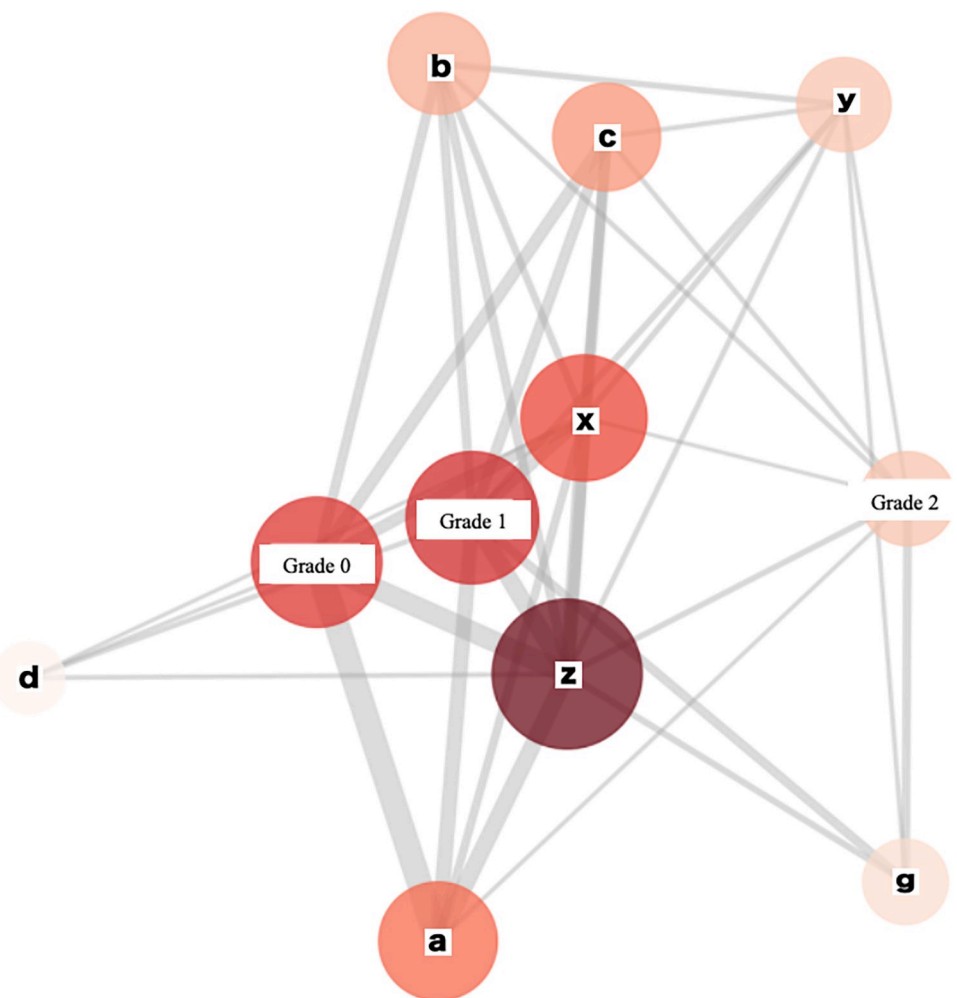

**Fig 22. Co-occurrence network diagram.** The node symbols are represented as follows: a: normal blood pressure, b: normal high blood pressure, c: high blood pressure, d: type I hypertension, e: type II hypertension, f: type III hypertension, g: isolated systolic hypertension (ISH), x: obesity, y: diabetes, z: dyslipidemia.

the model and help in defining treatment guidelines. In elderly patients, who often exhibit multimorbidity and polypharmacy, treatment can be particularly challenging. Therefore, a causal inference approach using directed networks, like a Bayesian network, would be effective in interpreting non-communicable diseases (NCDs) such as diabetes mellitus, cardiovascular diseases, cerebrovascular diseases, and dementia. This approach can be implemented using large-scale medical data from personal health records (PHRs). Unfortunately, in this study, the number of patients was insufficient to demonstrate interpretability for Grades 3 and 4. However, in Grades 0–2, we were able to show the relationship between hypertension grade and white matter hyperintensities (WMH) using a co-occurrence network.

## Acknowledgments

We would like to thank to the staff at Shin-Takeo Hospital who provided valuable data to conduct this study.

## Author Contributions

**Conceptualization:** Noriaki Takemura, Yuya Shinkawa, Kazuo Ishii.

**Formal analysis:** Noriaki Takemura, Kazuo Ishii.

**Investigation:** Kazuo Ishii.

**Methodology:** Noriaki Takemura.

**Project administration:** Kazuo Ishii.

**Resources:** Yuya Shinkawa.

**Software:** Noriaki Takemura, Kazuo Ishii.

**Supervision:** Kazuo Ishii.

**Validation:** Noriaki Takemura, Kazuo Ishii.

**Visualization:** Noriaki Takemura, Kazuo Ishii.

**Writing – original draft:** Noriaki Takemura, Kazuo Ishii.

**Writing – review & editing:** Noriaki Takemura, Kazuo Ishii.

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
