## [Decision Letter · Decision Letter 0]

14 Oct 2024

PONE-D-24-21793Grade prediction of lesions in cerebral white matter using a convolutional neural networkPLOS ONE

Dear Dr. Ishii,

Thank you for submitting your manuscript to PLOS ONE. After careful consideration, we feel that it has merit but does not fully meet PLOS ONE’s publication criteria as it currently stands. Therefore, we invite you to submit a revised version of the manuscript that addresses the points raised during the review process.

We look forward to receiving your revised manuscript.

Kind regards,

Xiaohui Zhang

Academic Editor

PLOS ONE

Reviewers' comments:

Reviewer's Responses to Questions

**Comments to the Author**

1. Is the manuscript technically sound, and do the data support the conclusions?

Reviewer #1: Yes

Reviewer #2: Partly

2. Has the statistical analysis been performed appropriately and rigorously? 

Reviewer #1: Yes

Reviewer #2: Yes

3. Have the authors made all data underlying the findings in their manuscript fully available?

Reviewer #1: Yes

Reviewer #2: Yes

4. Is the manuscript presented in an intelligible fashion and written in standard English?

Reviewer #1: Yes

Reviewer #2: Yes

5. Review Comments to the Author

Reviewer #1: The paper addresses the important topic of automating the prediction of cerebral white matter lesion grades using convolutional neural networks (CNN) and MRI data, which is both timely and relevant. The focus on integrating clinical data, such as hypertension, with MRI images to predict lesion severity is novel and could have significant clinical applications, especially in neuroimaging and diagnostic tools for cerebrovascular diseases. The authors used a comprehensive set of MRI modalities, including T1-weighted, T2-weighted, and FLAIR images, which provided robust input data for the CNN. The study's attempt to optimize image sizes and axes range enhances the precision of the model's performance, resulting in high test accuracy and area under the curve (AUC) scores for different lesion grades.

Comments:

(1) While the model performs well for grades 0 to 2, the small sample size for grades 3 and 4 (39 and 8 patients, respectively) weakens the generalizability of the results. This imbalance could lead to overfitting, particularly for higher lesion grades, as the model may not have learned enough representative features. Can the authors discuss about this limitation?

(2) Can the authors discuss about the generalizability of the proposed model for other populations or MRI devices?

(3) Can the authors give some interpretability of the model such as which MRI features or regions contribute most to the model's decisions would increase the clinical utility of the approach in the future?

Reviewer #2: In this paper, Takemura et al. established a diagnostic method for cerebral white matter lesions using MRI images and examined the relationship between the MRI images and the medical

checkup data using a co-occurrence network diagram.

The authors performed detailed evaluations of the performance of the convolutional neural network for the image size, range, and axial position of the MRI images. However, I found many details regarding the neural network experiments that need to be included, and the results lack solid explanation. I recommend that authors to address the concerns below:

Major comments:

1. Line 143-145: “Cerebral white matter lesions were evaluated using MRI images categorized into the following five grades.” what quantitative criteria are used to categorize the grades? Are there specific thresholds? Please expand the dividing methods here.

2. Fig 5:

a) It is unclear what the neural network’s output is in the paper or how the output images from the neural network are further used to classify the grades of white matter lesions.

b) How is the convolutional neural network created? What are the training data and testing data? And what is the loss function used in training? How did the author handle the imbalance in the number of MRI images across different grades?

3. Line 245-253: the definition of the image size is not clear here. Does the author downsample the original image or crop the original image to the target image size?

Minor comments:

1. I recommend using arrows to indicate cerebral white matter lesions in the image. This will help the reader see the regions more easily.

2. Line 329: “Figure 18”, a typo; the authors seem to refer to figure 19.

3. Fig 22: the letters inside each circle are too small

6. PLOS authors have the option to publish the peer review history of their article (what does this mean?). If published, this will include your full peer review and any attached files.

Reviewer #1: No

Reviewer #2: No

---

## [Author Response · Author response to Decision Letter 0]

16 Oct 2024

Comments of Reviewer #1: 

(1) While the model performs well for grades 0 to 2, the small sample size for grades 3 and 4 (39 and 8 patients, respectively) weakens the generalizability of the results. This imbalance could lead to overfitting, particularly for higher lesion grades, as the model may not have learned enough representative features. Can the authors discuss about this limitation?

Answer: 

I added Table 2 to clarify the quantitative criteria to categorize the grade. Results showed Grade distribution as Grade 0: Grade 1: Grade 2: Grade 3: Grade 4 = 551(48.1%): 441(38.5%): 107(9.3%): 39(3.4%): 8(0.7%) in 1146 subjects. That probably reflected real Grade distribution in this hospital and this region. Normally small number of patients shows high Grade variance in the grading. So, the results of Grade 3 and Grade 4 might show the less generalizability because of overfitting, particularly for higher lesion grades. Therefore, there may be a limitation in severe Grades in this Grading model. However, this model is enough practical to conduct in clinical use for screening, because physician can check and diagnose by details inspection. It will be able to be checked and diagnosed by details inspection of patients in such distribution and such number of patients of Grade 3 and Grade 4. If the number of cases the patient will accumulated, the model will be able to get more generalizability. So, this model should be updated to maintain the grading quality for keeping the practical generalizability. Because machine leaning model is data-driven, the model should maintain the grading quality and its generalizability by update the model using additive data and adequate data administration.

(2) Can the authors discuss about the generalizability of the proposed model for other populations or MRI devices?

Answer: 

As I mentioned in the previous Q & A, machine learning is normally data-driven equipment. So, the proposed model may be practical in clinical use for other population and MRI devices. However, there is some chips for keeping generalizability because of the data-driven equipment. The grading quality should be maintained by updating using the adequate training data because grading quality is depending on the training data. To apply the proposed model to other populations or MRI devices, the ML user should keep and maintain the grading quality using some techniques such as transfer learning and fine tuning and should evaluate data grading quality. So, the proposed model is not final and not perfect for generalizability. The model should be maintained and updated for keeping practical grading quality.

(3) Can the authors give some interpretability of the model such as which MRI features or regions contribute most to the model's decisions would increase the clinical utility of the approach in the future?

Answer: 

Yes, we can add some interpretability of the model using causal inference approach such as logistic regression analysis and bayesian network model instead of co-occurrence network. That is a directed network instead of an undirected network. Using directed network, we can give some interpretability of the model and show the treatment guidelines. In elderly, the patients show multimorbidity and polypharmacy, and it is sometimes very difficult to treat. So, the causal inference approach using directed network, such as bayesian network, should be effective for interpretability of non-communicable diseases (NCDs), such as diabetes mellitus, cardiovascular diseases, cerebrovascular diseases and dementia. Using the medical big data PHRs, we can realize this approach. Unfortunately, in this study, the patient number is not so enough to show the interpretability in Grade 3 and Grade 4. But in Grade 0 - 2, we could show the relationship between hypertension grade and WMH using co-occurrence network.

According to above answers, I added the below sentences into the discussion:

The grade distribution was as follows: Grade 0: 551 (48.1%), Grade 1: 441 (38.5%), Grade 2: 107 (9.3%), Grade 3: 39 (3.4%), and Grade 4: 8 (0.7%) among 1,146 subjects. This likely reflects the actual grade distribution in this hospital and region. Normally, a small number of patients show high grade variance in the grading. The results for Grades 3 and 4 may indicate less generalizability, probably due to overfitting. Therefore, there may be some limitations in the higher grades within this grading model. However, the model might still be practical for clinical use in screening, as physicians can perform detailed inspections to verify diagnoses. As the number of cases accumulates, the model should be able to achieve greater generalizability. 

Since machine learning is typically data-driven, the proposed model may be practical for clinical use across different populations and MRI devices. However, there are some challenges in maintaining generalizability. The grading quality should be upheld by continuously updating the model with an appropriate training dataset, as the quality of grading depends on the quality of the training data. To apply the proposed model to different populations or MRI devices, ML users must ensure and maintain grading quality through techniques like transfer learning and fine-tuning, while also evaluating the quality of the data grading. Therefore, the proposed model is neither final nor perfect in terms of generalizability.

We can enhance the model's interpretability by using causal inference approaches, such as logistic regression analysis and Bayesian network models, instead of a co-occurrence network. A directed network, as opposed to an undirected network, can provide some interpretability to the model and help in defining treatment guidelines. In elderly patients, who often exhibit multimorbidity and polypharmacy, treatment can be particularly challenging. Therefore, a causal inference approach using directed networks, like a Bayesian network, would be effective in interpreting non-communicable diseases (NCDs) such as diabetes mellitus, cardiovascular diseases, cerebrovascular diseases, and dementia. This approach can be implemented using large-scale medical data from personal health records (PHRs). Unfortunately, in this study, the number of patients was insufficient to demonstrate interpretability for Grades 3 and 4. However, in Grades 0-2, we were able to show the relationship between hypertension grade and white matter hyperintensities (WMH) using a co-occurrence network. 

Comments of Reviewer #2:

Major comments:

1. Line 143-145: “Cerebral white matter lesions were evaluated using MRI images categorized into the following five grades.” what quantitative criteria are used to categorize the grades? Are there specific thresholds? Please expand the dividing methods here.

Answer:

I added Table 2 as shown below:

Cerebral white matter lesions were evaluated using MRI images categorized into the following five grades: grade 0, in which no lesions are detected through grade 4, in which most lesions are observed [13], as shown in Table 2 [4,14]. 

Table 2. Grading the severity of white matter hyperintensities (WMH) [4,14]

Deep and subcortical white matter hyperintensity; DSWMH

Grade 0 Absence

Grade 1 Punctuate foci (< 3 mm in diameter)

Grade 2 Punctuate foci (≥ 3 mm in diameter)

Grade 3 Confluence of foci with unclear boundary

Grade 4 Large confluent areas

And I added Reference 4 of Journal data: Cerebrovasc Dis. 2007;24(2-3):202-9.

Major comments:

2. Fig 5:

a) It is unclear what the neural network’s output is in the paper or how the output images from the neural network are further used to classify the grades of white matter lesions.

Answer:

The neural network’s output consists of probabilities for Grade 0, Grade 1, Grade 2, Grade 3, and Grade 4 using the output layer (type=Dense, output=5, activation=softmax). The grade with the highest prediction probability was classified as the predicted grade.

b) How is the convolutional neural network created? What are the training data and testing data? And what is the loss function used in training? How did the author handle the imbalance in the number of MRI images across different grades?

Answer:

How is the convolutional neural network created?:

The convolutional neural network was constructed using Keras module (Sequential from keras.models, Conv2D (Convolution layer), MaxPooling2D (Pooling layer), Activation, Dropout (Dropout layer), Flatten (Flatten layer), and Dense (Dense layer) from keras.layers). It consists of thirteen layers, as shown in Figure 5 and Table 5, with the following structure: Convolution, Convolution, Pooling, Dropout, Convolution, Convolution, Pooling, Flatten, Dense, Dropout, Dense, Dropout, Dense. 

What are the training data and testing data?:

Each learning process was performed by mini batch method (batch size=32). And the training data and testing data were constructed automatically by model_selection.training_test_split method of sklearn module. Each batch was automatically and randomly split into (training data: testing data)=3:1.

And what is the loss function used in training?:

The loss function used in training is categorical_crossentropy from the Keras module. The neural network's output provides probabilities for Grade 0, Grade 1, Grade 2, Grade 3, and Grade 4, using the output layer (type=Dense, output=5, activation=softmax). The grade with the highest prediction probability is classified as the predicted grade.

How did the author handle the imbalance in the number of MRI images across different grades?

Each grade evaluation in ROC curve was conducted using a dataset consists of (indicated Grade: other Grades)=1:1, e.g. (Grade0: Grade1-4)=1:1.

Major comments:

3. Line 245-253: the definition of the image size is not clear here. Does the author downsample the original image or crop the original image to the target image size?

Answer:

The original MRI image was changed from 30x30 to 300x300 pixels using the Image.resize() method of the Python Imaging Library PIL, and analyzed.

According to above answers, I added explanations into Methods and Results:

Methods:

The convolutional neural network was constructed using the Keras module, specifically Sequential from keras.models, and Conv2D (convolution layer), MaxPooling2D (pooling layer), Activation, Dropout (dropout layer), Flatten (flatten layer), and Dense (dense layer) from keras.layers. It consists of thirteen layers, as shown in Figure 5 and Table 5, with the following structure: Convolution, Convolution, Pooling, Dropout, Convolution, Convolution, Pooling, Flatten, Dense, Dropout, Dense, Dropout, Dense. Each learning process was performed using the mini-batch method (batch size = 32). The training and testing data were automatically generated by the model_selection.train_test_split method from the sklearn module. Each batch was randomly split into a 3:1 ratio for training data and testing data, respectively. The loss function used in training is categorical_crossentropy from the Keras module. The neural network’s output provides probabilities for Grade 0, Grade 1, Grade 2, Grade 3, and Grade 4 through the output layer (type=Dense, output=5, activation=softmax). The grade with the highest predicted probability is classified as the predicted grade. Each grade evaluation in the ROC curve was conducted using a dataset consisting of a 1:1 ratio between the indicated grade and the other grades, e.g., (Grade 0: Grades 1-4)=1:1.

Results:

The original MRI image was changed from 30x30 to 300x300 pixels using the Image.resize() method of the Python Imaging Library PIL, and analyzed.

Minor comments:

1. I recommend using arrows to indicate cerebral white matter lesions in the image. This will help the reader see the regions more easily.

Arrows were added into Fig 2 and Fig 3.

2. Line 329: “Figure 18”, a typo; the authors seem to refer to figure 19.

Answer:

Thank you for your suggestion. I added Fig 18 because I forget to add Figure 18.

3. Fig 22: the letters inside each circle are too small

Answer:

Fig 22 was modified with large letters.

---

## [Decision Letter · Decision Letter 1]

25 Oct 2024

Grade prediction of lesions in cerebral white matter using a convolutional neural network

PONE-D-24-21793R1

Dear Dr. Ishii,

We’re pleased to inform you that your manuscript has been judged scientifically suitable for publication and will be formally accepted for publication once it meets all outstanding technical requirements.

Kind regards,

Xiaohui Zhang

Academic Editor

PLOS ONE

Additional Editor Comments (optional):

Reviewers' comments:

Reviewer's Responses to Questions

**Comments to the Author**

1. If the authors have adequately addressed your comments raised in a previous round of review and you feel that this manuscript is now acceptable for publication, you may indicate that here to bypass the “Comments to the Author” section, enter your conflict of interest statement in the “Confidential to Editor” section, and submit your "Accept" recommendation.

Reviewer #1: All comments have been addressed

Reviewer #2: All comments have been addressed

2. Is the manuscript technically sound, and do the data support the conclusions?

Reviewer #1: Yes

Reviewer #2: Yes

3. Has the statistical analysis been performed appropriately and rigorously? 

Reviewer #1: Yes

Reviewer #2: Yes

4. Have the authors made all data underlying the findings in their manuscript fully available?

Reviewer #1: Yes

Reviewer #2: Yes

5. Is the manuscript presented in an intelligible fashion and written in standard English?

Reviewer #1: Yes

Reviewer #2: Yes

6. Review Comments to the Author

Reviewer #1: (No Response)

Reviewer #2: (No Response)

7. PLOS authors have the option to publish the peer review history of their article (what does this mean?). If published, this will include your full peer review and any attached files.

Reviewer #1: No

Reviewer #2: No

---

## [Editor Report · Acceptance letter]

31 Oct 2024

PONE-D-24-21793R1 

PLOS ONE

Dear Dr. Ishii, 

I'm pleased to inform you that your manuscript has been deemed suitable for publication in PLOS ONE. Congratulations! Your manuscript is now being handed over to our production team.

Kind regards, 

on behalf of

Dr. Xiaohui Zhang 

Academic Editor

PLOS ONE